

# Aqueous-phase chemistry of glyoxal with multifunctional reduced nitrogen compound: A potential missing route of secondary brown carbon

Yuemeng Ji[1,2], Zhang Shi[1,2], Wenjian Li[1,2], Jiaxin Wang[1,2], Qiuju Shi[1,2], Yixin Li[3], Lei Gao[1,2], Ruize Ma[1,2], Weijun Lu[2], Lulu Xu[1,2], Yanpeng Gao[1,2], Guiying Li[1,2], Taicheng An[1,2]

[1]Guangdong Key Laboratory of Environmental Catalysis and Health Risk Control, Guangdong-Hong Kong-Macao Joint Laboratory for Contaminants Exposure and Health, Institute of Environmental Health and Pollution control, Guangdong University of Technology, Guangzhou 510006, China;
[2]Guangzhou Key Laboratory of Environmental Catalysis and Pollution Control, Key Laboratory of City Cluster Environmental Safety and Green Development, School of Environmental Science and Engineering, Guangdong University of Technology, Guangzhou 510006, China.
[3]Department of Chemistry, University of California Irvine, Irvine, CA 92697, USA

*Correspondence to*: Prof. Taicheng An (antc99@gdut.edu.cn)

**Abstract.** Aqueous-phase chemistry of glyoxal (GL) with reduced nitrogen compounds (RNCs) plays a significant source of secondary brown carbon (SBrC), which is one of the largest uncertainties in climate predictions. However, few studies have revealed that SBrC formation is affected by multifunctional RNCs, which has a non-negligible atmospheric abundance. Hence, we performed theoretical and experimental approaches to investigate the reaction mechanisms and kinetics of the mixtures for ammonium sulfate (AS), multifunctional amine (monoethanolamine, MEA) and GL. Our experiments indicate that the light-absorption and the growth rate are enhanced in MEA-GL mixture relative to AS-GL and MEA-AS-GL mixtures, and MEA reactions of the chromophores by more efficiently than the analogous AS reactions. Quantum chemical calculations show that the formation and propagation of oligomers proceed via four-step nucleophilic addition reactions in three reaction systems. The presence of MEA provides the extra two branched chains to affect the natural charges and steric hindrance of intermediates, facilitated the formation of chromophores. Molecule dynamics simulations reveal that the interfacial and interior attraction on the aqueous aerosols with MEA is more pronounced for small α-dicarbonyls, to facilitate the further engagement in the aqueous-phase reactions. Our results show a possible missing source for SBrC formation on urban, regional and global scales.

## 1 Introduction

Brown carbon (BrC) represents the most important source of carbonaceous aerosols, with profound implications to the global climate, air quality and human health (Laskin et al., 2015; Marrero-Ortiz et al., 2018; Li et al., 2022; Yan et al., 2018; Yuan et al., 2023). Chemical transport models reveal that a non-negligible radiative forcing by BrC is range from 0.05 to 0.27 W m$^{-2}$ averaged globally (Tuccella et al., 2020; Wang et al., 2018; De Haan et al., 2020; Zhang et al., 2020; Laskin et al., 2015; Moise et al., 2015). Large differences in these estimated data result from the uncertainties of BrC on its formation mechanisms,



chemical composition and optical properties (An et al., 2019; Shi et al., 2020; Kasthuriarachchi et al., 2020; Corbin et al., 2019). It affects understanding the radiative effect in current climate models (Liu et al., 2020; Zhang et al., 2020; Zhang et al., 2023). Compared with primary BrC, sources and formation of secondary BrC (SBrC) are more complex and lack of understanding in detail (Lin et al., 2015; Yuan et al., 2020; Srivastava et al., 2022). Hence, in recent years, great efforts have been made to better understand the chemical composition and formation mechanisms of SBrC chromophores.

There is compelling evidence that the heterogeneous reactions of reduced nitrogen compounds (RNCs) and small α-dicarbonyls have been recognized as significant sources of SBrC (Hawkins et al., 2018; De Haan et al., 2018; George et al., 2015). These SBrC chromophores are normally conjugated and possibly heteroaromatic species, such as imidazole (IML) and its derivatives (De Haan et al., 2009b; De Haan et al., 2009a; Yang et al., 2022). Numerous previous studies paid much attention to BrC from the secondary processes of small α-dicarbonyls with ammonium sulfate (AS) and methylamine (MA) (De Haan et al., 2020; De Haan et al., 2019; De Haan et al., 2009a; Lin et al., 2015). For example, nearly 30 chromophores were detected in AS-methylglyoxal (MG) mixture by HPLC/PDA/HRMS and nitrogen-containing compounds account for more than 70% of the overall light absorption within 300−500 nm range (Lin et al., 2015). Some studies have also revealed that the absorption of BrC generated in AS- or MA-MG mixture increases with pH value (Hawkins et al., 2018; Sedehi et al., 2013) Also, the iminium pathway is predominant while pH < 4 to form IML and its derivatives but is suppressed at pH 4.(Nozière et al., 2009; Sedehi et al., 2013; Yu et al., 2011). Hence, pH value has a large effect on the formation of SBrC chromophores, but the chemical mechanisms of BrC formation under the different pH values remain unclear, hindering a systematical understanding its integrated atmospheric chemistry and nonnegligible environmental impacts.

On the other hand, multifunctional RNCs (such as ethanolamines and amino acids) display a strong atmospheric activity to the formation of SBrC with an unneglected atmospheric concentration (Huang et al., 2016; Ge et al., 2011; Powelson et al., 2014; Trainic et al., 2012; Laskin et al., 2015; Ning et al., 2022). For example, a rapid BrC formation was detected in glycine reactions with small α-dicarbonyls, and sub-micrometer amino acids particles exhibited a high growth upon exposure to small α-dicarbonyls (Powelson et al., 2014; Sedehi et al., 2013; De Haan et al., 2009b; Trainic et al., 2012). On the other hand, monoethanolamine (MEA) is an amine-based solvent for post-combustion $CO_2$ capture (PCCC) technology with a relatively high vapor pressure, emitting 80 tons per year into the atmosphere for each 1 million tons of $CO_2$ removed per year (Karl et al., 2011; Puxty et al., 2009; Shen et al., 2019). Recent field measurement has shown that MEA is the second most abundant organic amine in $PM_{2.5}$ in Shanghai besides MA (Huang et al., 2016). However, to the best of our knowledge, few previous results are available on the participation of MEA in the SBrC formation with small α-dicarbonyls and its potential role in the atmosphere and human health were also not attempted.

Hence, we elucidated the chemical mechanisms of BrC chromophores from the mixtures of typical reaction of RCNs (i.e., MEA and AS) with small α-dicarbonyls using combined theoretical and experimental methods. Herein, glyoxal (GL) is selected as the representative of small α-dicarbonyls due to its high global emissions and significant contribution to BrC (Fu et al., 2008; Myriokefalitakis et al., 2008; Shi et al., 2020; Nie et al., 2022; Gomez et al., 2015). The chemical composition of the BrC chromophores was characterized by mass spectrometry in different initial pH values, and the optical properties were



measured using UV-Vis spectrophotometry. Possible pathways were calculated using density functional theory, and the
mechanism of BrC chromophore formation was also simulated. The effects of multifunctional amines in formation of SBrC
chromophores were elaborated further. Additionally, the potential implications of multifunctional amines on climate radiative
forcing were stated and discussed briefly.
**2 Experimental methods and theoretical calculations**
**2.1 Experimental section**
The procedures of each experiment are summarized in Fig. S1. All reagents were used as described in Supporting Information
(SI). Three mixtures were prepared under atmospheric relevant aqueous conditions to generate SBrC: AS-GL, MEA-GL and
MEA-AS-GL. Briefly, AS-GL (1 M) mixture was prepared by adding AS to aqueous GL (in ultrapure water) for a final
concentration of 1 M of each reactant in the volumetric flasks. For the two MEA-containing mixtures, MEA was acidified with
diluted sulfuric acid (20%) to prevent GL from reacting with MEA in alkaline condition. The acidified MEA was then combined
with aqueous GL similar to that described for the AS-GL (1 M) mixture. All three solutions mentioned above were then diluted
to reach a final concentration of 1 M in three 50mL volumetric flasks. To explore the effects of pH values, three mixtures were
prepared with an initial pH values of 3 or 4 via addition of sulfuric acid (20%) or sodium hydroxide solution (2 M) prior to the
mixing of RNCs and GL (Kampf et al., 2016; Yu et al., 2011). Each mixture was then transported into brown vials for 15 days
in dark condition, which has been proved efficient formation of chromophores in droplet evaporation collecting on the
timescales of seconds (Zhao et al., 2015; Lee et al., 2014).
The absorption spectrums of all mixtures were recorded by using an UV-Vis spectrophotometer (Agilent Cary 300, USA).
All experimental solutions were diluted by a factor of 200 or 400 before each measurement to avoid saturation of the absorption
peaks. The diluted samples were added into a quartz cuvette with 1 cm optical path length right away to prevent the diluted
samples from photolysis. The spectrums recorded between 200 – 500 nm were shown in Fig. 1. And the blank experiments of
GL and RNCs solution were performed and presented in Fig. S2. The absorption spectrums of all samples were measured with
three times. The wavelength-dependent mass absorption coefficients (MACs) of experimental solutions were calculated from
initial base-10 absorbance ($A_{10}$),
$$\text{MAC}(\lambda) = \frac{A_{10}^{\text{solution}}(\lambda) \times \ln(10)}{b \times C_{\text{mass}}}$$

where $C_{\text{mass}}$ is the mass concentration of reactants and $b$ is path length (Aiona et al., 2017; Chen and Bond, 2010). The different
dilution factors were normalized by using MAC formula.
Samples used for mass spectrometry analysis were diluted by a factor of 800 or 1000 followed by syringe filtration. The
filters were stored in brown chromatography injection vials to block the light. Ultra-performance liquid chromatography
coupled to hybrid Quadrupole-Exactive Orbitrap mass spectrometry (UPLC-Q-Orbitrap HRMS, Thermo Scientific™, USA)
(Wang et al., 2017) was employed to obtain structural data of chromophores in this study. $MS^2$ analysis were used for all



chromophores with a weight error of less than 10 ppm compared with the theoretical mass to obtain fragments information for
the identification of structure analysis. Detailed description of the mass spectrometry and chromatographic conditions are all
described in SI.

**2.2 Quantum calculations and molecular dynamics simulations**

Quantum chemical calculations were performed using the Gaussian 09 package (M. J.Frisch, 2013). Structures for all
stationary points (SPs), including reactants, intermediates, transient states (TSs), and products, were optimized using the hybrid
density functional of M06-2X method(Zhao and Truhlar, 2007) with 6-311G(d,p) basis set, i.e., at the M06-2X/6-311G(d,p)
level (Ji et al., 2017). The solvent effect was considered using the solvation model based on density (SMD) to simulate the
aqueous environment (Gao et al., 2016; Marenich et al., 2009). Harmonic frequency calculation was carried out at the same
level as structural optimization to verify whether SP is a TS (with one and only imaginary frequency) or a minimum (without
imaginary frequencies) (Ji et al., 2022). Intrinsic reaction coordinate calculation was performed to confirm that the TSs
connected with the corresponding reactants and products. Single point energy (SPE) calculation was executed using the M06-
2X method with a more flexible 6-311+G(3df,3pd) basis set to obtain more accurate potential energy surfaces (PESs). For the
pathways with TSs, the rate constants ($k$) were calculated via conventional transition state theory (TST) (Evans and Polanyi,
1935; Eyring, 1935; Galano and Alvarez-Idaboy, 2009; Gao et al., 2014). To simulate real atmospheric conditions in the
solution, the calculated $k$ values were refined by solvent cage effects(Okuno, 1997) and diffusion-limited effects (Collins and
Kimball, 1949), of which the calculation details of diffusion-limited rate constant $k_d$ can be seen in SI. For the pathways without
TSs, the corresponding $k$ values are predominated by the diffusion-limit effect which equal to the diffusion-limited rate
constants.
Classical molecular dynamics (MD) was performed using NAMD package (Phillips et al., 2005) to simulate the
heterogeneous processes of GL from gas to the AS and MEA particles. The AS particle is composed of 39 $SO_4^{2-}$, 78 $NH_4^+$ and
2046 $H_2O$ in a box size of $40 \times 40 \times 40$ Å$^3$, while the MEA particle consists of 39 MEA and 2036 $H_2O$. The 5 ns equilibration
at the time step of 1 fs was executed in the isothermal-isochoric ($NVT$) ensemble ($T$ = 298 K) to ensure the thermodynamic
equilibrium of particles (Shi et al., 2020; Zhang et al., 2019). The MD simulation of 2 ns is run via the $NVT$ ensemble. MEA
and GL were described using CHARMM force field (Jorgensen et al., 1996), and $H_2O$ using TIP3P model (Martins-Costa et
al., 2012). The fixed charges on $NH_4^+$ and $SO_4^{2-}$ are scaled by 0.75 to account for the electronic polarizability (Leontyev and
Stuchebrukhov, 2011; Mosallanejad et al., 2020). The periodic boundary conditions were selected for three dimensions. In
order to calculate the kinetic trajectories of GL from gas to two target particles, the free energy profile along the distance of
the center of mass between each particle and GL was calculated via umbrella sampling (Torrie and Valleau, 1977) and
weighted-histogram analysis method (Kumar et al., 1992) based on the above equilibrated molecular dynamics trajectories.
The bias potential force constant was equal to 10 kcal mol$^{-1}$ Å$^{-2}$.





## 3 RESULTS AND DISCUSSION

### 3.1 Mass absorption coefficients of BrC chromophores

The mass absorption coefficients (MACs) identified in AS-GL, MEA-GL, and MEA-AS-GL mixtures at the initial pH of 3 and 4 (denoted as pH = 3 and pH = 4) are shown in Fig. S3. The maximum adsorption peaks locate at 207, 212, and 209 nm for AS-GL, MEA-GL, and MEA-AS-GL mixtures at pH = 3, respectively, which are the characteristic of BrC chromophores. Also the corresponding location is not changed at pH = 4. The MAC values of the maximum adsorption peaks are in the range of $1080-17909$ $cm^2$ $g^{-1}$ for three mixtures, which are quantitatively similar to other aerosol samples generated in laboratory smog chambers with MACs of SBrC on the order of $\sim 10^{2-4}$ $cm^2$ $g^{-1}$ (Ackendorf et al., 2017; Kampf et al., 2016; Lee et al., 2013; Powelson et al., 2014; Shapiro et al., 2009; Yu et al., 2011; Zhao et al., 2015). The MAC values at pH = 4 are higher than those at pH = 3 for three mixtures. For example, the MAC value in AS-GL mixture is 2037 $cm^2$ $g^{-1}$ at pH = 4, which is almost twice higher than that at pH = 3. Hence, the initial pH values of solution mainly affect the MAC values rather than the locations of absorption peaks.

Fig. 1a shows a comparison of the MAC values of all three mixtures at the initial pH of 3 and 4. The MAC values of maximum adsorption peaks increase from AS-GL to MEA-GL to MEA-AS-GL mixture, ranging from 1080 to 6345 $cm^2$ $g^{-1}$ at pH = 3 and 2037 to 7617 $cm^2$ $g^{-1}$ at pH = 4. The highest MAC value of MEA-AS-GL is explained by the different initial total concentration of reactants (see in Method), since the initial concentration of AS and MEA in MEA-AS-GL mixture is twice times than that in MEA-GL or AS-GL mixture. In addition, the MAC value of maximum adsorption peak in MEA-AS-GL mixture is higher than the sum of those in MEA-GL and AS-GL mixtures, and the location of maximum absorption peak in MEA-AS-GL mixture is between those in MEA-GL and AS-GL mixtures. It implies that the extra chromophores are yielded in MEA-AS-GL mixture in addition to producing the same chromophores as AS-GL and MEA-GL mixtures.

To compare the formation rate of chromophore between the different mixtures, the growth rates (GRs) of the maximum absorption peaks as a function of reaction time is shown in Fig. 2. The trend of the GR variation with reaction time at pH = 3 is similar to that at pH = 4, while the GRs of three mixtures at pH = 4 are larger than those at pH = 3 at the beginning of the reactions. The GRs are nearly invariant after $6-9$ days, implying that the chromophore formation for three mixtures is irreversible. MEA-AS-GL mixture exhibits the larger GRs than other mixtures at the beginning of reaction because of its higher initial concentration of reactants. As the reaction proceeds, the GRs of MEA-GL mixture are increased and finally larger than those of other mixtures. Hence, MEA reactions form the chromophores by more efficiently than the analogous AS reactions.

The GRs dependence of the pH values of three mixtures is also plotted as a function of reaction time as shown in Fig. 2. The pH values rapidly degrade within the first 2 days in three mixtures, which is the same trend as GRs that decrease by a factor of more than $1-3$ at pH = 3 and 4. This trend is explained by ambient pH values, since a known byproduct (i.e., formic acid) is formed (De Haan et al., 2009b; De Haan et al., 2020; Galloway et al., 2009; Hamilton et al., 2013; Kampf et al., 2012; Yu et al., 2011). Note that the trend of GRs shows a decrease from MEA-AS-GL, MEA-GL, to AS-GL mixtures at the beginning of the reaction time, while the MAC values of MEA-GL mixture are larger than those of two mixtures because of the smaller



pH value in solution after the reaction is equilibrium (Figs. 1b and 2), suggesting that chromophore formation of three mixtures
depends on the ambient pH value.

### 3.2 Chemical composition characterization of BrC chromophores

The chemical composition characterization of formed BrC chromophore were conducted by UPLC-Q-Orbitrap HRMS. The
formulas, m/z values, characteristic fragments, and structures of chromophores and intermediates are identified based on
obtained mass spectrum data in AS-GL, MEA-GL, and MEA-AS -GL mixtures (Table S1). The corresponding MS and $MS^2$
spectrums of chromophores and intermediates are exhibited in Fig. 3 and Figs. S4-S8. For all mixtures, imidazole (IML)
compounds are identified with a characteristic peak at m/z 69.045 in $MS^2$ spectrums. Therefore, various IML compounds are
observed based on several representative peaks at m/z 69.045, including imidazole ($IML_{AS}$ and $IML_{MEA}$), imidazole-2-
carboxaldehyde ($IC_{AS}$ and $IC_{MEA}$), and their hydrated forms ($HIC_{AS}$ and $HIC_{MEA}$) for AS-GL and MEA-GL mixtures (Table
S1, Figs. 3a-b and S4-S5). For MEA-GL mixture, extra catenulate intermediates without IML-structure characteristics are
obtained at m/z values of 102.055 and 120.065 (Table S1, Figs. 3a and S6), corresponding to $C_4H_7O_2N$ ($IA_{MEA}$) and $C_4H_9O_3N$
($AHA_{MEA}$ and $ID_{MEA}$) compounds, respectively. However, no catenulate intermediates in AS-GL mixture are observed in this
study because of their low concentrations and short lifetimes, although they are observed by previous studies using MS/AMS
and $^1H$ nuclear magnetic resonance spectroscopy (Galloway et al., 2009; Lee et al., 2013; Yu et al., 2011). In addition, as shown
in Fig. 3b and Fig. S7, some IML-based products at m/z values of 145.061, 135.066, and 193.072 were obtained in AS-GL
mixture correspond to hydrated N-glyoxal substituted imidazole ($HGI_{AS}$), 2,2'-biimidazole ($BIM_{AS}$), and its glyoxal substituted
analog ($GBI_{AS}$), respectively. As discussed above, an important distinction between AS-GL and MEA-GL mixtures is whether
formation of bicyclic IML products (Fig. 3a-b), indicating that the optical properties of chromophores are mainly determined
by mono-imidazole compounds rather than bicyclic IML compounds.
To further explore the difference of identified products in MEA-GL and AS-GL mixtures, the possible pathways leading
to the identified intermediates and chromophores are illustrated in Fig. 4, along with the reaction energies ($\Delta G_r$) of all pathways
calculated at the M06-2X/6-311+G(3df,3pd)//M06-2X/6-311G(d,p) level. As shown in Fig. 4, the formation and propagation
of oligomers was proposed to proceed via four-step nucleophilic addition (NA) reactions. For MEA-GL mixture, three
catenulate intermediates ($AHA_{MEA}$, $IA_{MEA}$, and $ID_{MEA}$) are successively yielded by the nucleophilic attack of MEA at the
reactive carbonyl site via dehydration and hydration, with the total $\Delta G_r$ value of $-7.8$ kcal mol$^{-1}$ (Fig. 4a). Subsequently, two-
step NA reactions between $ID_{MEA}$ and MEA and between $DI_{MEA}$ and GL-diol (DL), followed by protonation and dehydration,
yields two intermediates ($HA_{MEA}$ and $PIC_{MEA}$) in sequence. Although the third NA reaction between $DI_{MEA}$ and DL is
endothermic ($\Delta G_r = 12.7$ kcal mol$^{-1}$), the total $\Delta G_r$ value of $DI_{MEA}$ formation in MEA-GL mixture is $-18.7$ kcal mol$^{-1}$ for
proceeding the NA reaction to yield $PIC_{MEA}$. Similarly, the formation of $PIC_{AS}$ in AS-GL mixture is also thermodynamically
feasible, with the total $\Delta G_r$ value of $-10.9$ kcal mol$^{-1}$. However, $PIC_{MEA}$ or $PIC_{AS}$ is thermodynamically unstable, since there
is a large exothermicity of the subsequent reaction pathway ($\Delta G_r = -78.6$ or $-50.0$ kcal mol$^{-1}$) for proceeding cyclization
leading to the formation of $IC_{MEA}$ or $IC_{AS}$. It should be noted that for AS-GL mixture, the fate of $IC_{AS}$ is dependent of the



competition between the pathways of hydration to yield $HIC_{AS}$ and NA reaction with DL to form $BI_{AS}$, while for MEA-GL
mixture, there are no nucleophilic sites of $IC_{MEA}$ for further oligomerization to form bicyclic IML compounds because $IC_{MEA}$
is imidazolium cation. Current results further explain our experimental results mentioned above that higher MAC and larger
GR values in MEA-GL mixture than that in AS-GL mixture.

198         For MEA-AS-GL mixture, the products in AS-GL and MEA-GL mixtures are also observed (Fig. 3c). Beyond that, four

extra IML compounds are also observed at m/z values of 113.071, 141.066, 159.076 and 171.076, corresponding to IML
($IML_{MAG}$), imidazole-2-carboxaldehyde ($IC_{MAG}$) and its hydrated form ($HIC_{MAG}$), and N-glyoxal substituted imidazole ($GI_{MAG}$)
(Fig. 3c and Fig. S8). An extra -$C_2H_4O$ group exists in the geometries of the above four IML compounds relative to the products
of AL-GL mixture, indicating that there exist the cross reactions between MEA and AS in the MEA-AS-GL mixture. As shown
in Fig. 5, the cross NA reaction between $ID_{AS}$ and MEA or $ID_{MEA}$ and AS possesses a negative $\Delta G_r$ value of −4.8 or −5.4 kcal
$mol^{-1}$, followed by dehydration to form the same intermediate, diimine ($DI_{MAG}$). It implies that the cross reactions in MEA-
AS-GL mixture are thermodynamical favorable. Therefore, the formation and propagation of chromophores in MEA-AS-GL
mixture also proceed via NA reactions, which is the key route for the formation of BrC chromophores.

207         As shown in Fig. 3c, no bicyclic IML compounds are produced in MEA-AS-GL mixture because the precursors of bicyclic

IML compound (i.e., imidazole-2-carboxaldehyde) is fully hydrated under more acidic condition than AS-GL mixture (see pH
values in Table S2). It leads to the formation of N-glyoxal substituted imidazole (i.e., $GI_{MAG}$) instead of bicyclic IML
compounds. The similar phenomenon is also found in the previous studies (Ackendorf et al., 2017; Kampf et al., 2012; Yu et
al., 2011) that bicyclic IML compounds are hardly yield from imidazole-2-carboxaldehyde in acidic condition. As discussed
above, imidazole-based structural characteristics in chromophores are maintained in the presence of MEA, but the
nucleophilicity of chromophores is reduced because the nucleophilic sites are occupied. Also, the positively charged quaternary
amine salts (such as $IC_{MEA}$ and $GI_{MAG}$) are also yield in MEA-GL and MEA-AS-GL mixtures, and thereby the chemical
composition and optical properties of chromophores are affected.
**3.3 Chemical reaction mechanism leading to BrC chromophores**
As discussed above, the four-step NA reactions are the key pathways to form and propagate oligomers including intermediates
and chromophores for three mixtures. Therefore, all possible pathways involved in the four key NA reactions of three mixtures
are calculated using density functional theory. The corresponding PESs stablished by the M06-2X/6-311+G(3df,3pd)//M06-
2X/6-311G(d,p) level are also presented in dotted boxes of Figs. 4-5. The optimized geometries of key stationary points,
including transition states (TSs), intermediates, and products, are depicted in Figs. S9-S11 at the M06-2X/6-311G(d,p) level.
We first performed quantum chemistry calculation to evaluated the direct nucleophilic attack of GL by MEA or AS, which
proceeds a large activation energy ($\Delta G^{\ddagger}$) value of 6.3 or 8.6 kcal $mol^{-1}$, following by H-shift reaction to yield $AHA_{MEA}$ or
$AHA_{AS}$, with also a large $\Delta G^{\ddagger}$ value of 15.2 or 18.2 kcal $mol^{-1}$ (see NA1a' and NA2a' in Fig. 4). The high $\Delta G^{\ddagger}$ values and
large endothermicity of the direct NA reactions leading to $AHA_{MEA}$ and $AHA_{AS}$ imply that their occurrences are kinetically
and thermodynamically hindered.



Hence, we explored the cationic oligomerization of chromophore formation under acidic condition, which involves three
essential steps, (1) protonation and dehydration to form cationic intermediates (CIs) or carbenium ions (CBs), (2) nucleophilic
attack of CIs or CBs by MEA and AS, and (3) formation of intermediates and chromophores by deprotonation or dehydration.
As shown in Figs. 4-5, each pathway involved in the cationic-mediated reaction mechanism proceeds without a TS, except
deprotonation of CIs, in line with the results of the previous studies (Ji et al., 2020; Ji et al., 2022). However, deprotonation of
CIs by sulfate ion ($SO_4^{2-}$) possesses a negative $\Delta G^{\ddagger}$ value in this study, implying an approximate barrierless process of this
kind of deprotonation.
For the first-step NA reaction (NA1a in Fig. 4) in MEA-GL mixture, the electrophilic cationic site of $CB_{DL}$ is attacked by
the nucleophilic -$NH_2$ group of MEA with the $\Delta G_r$ value of $-40.3$ kcal mol$^{-1}$. $CB_{DL}$ is broadly produced from GL and reflected
from the large particle growth and formation of IML products (Ji et al., 2020; Li et al., 2021). The deprotonation of $CI_{MEA}1$
possesses a negative $\Delta G^{\ddagger}$ value of $-4.5$ kcal mol$^{-1}$, and a pre-reactive complex is identified prior the corresponding TS (detailed
in SI). Similarly, the other two NA1b and NA1c reactions (Fig. 4) also include protonation, dehydration, nucleophilic attack,
and deprotonation to yield $HA_{MEA}$ and $PIC_{MEA}$. Kinetic data listed in Table S3 show that the rate constants of most pathways
involved in the NA1a-1b and NA2a-2c reactions fall in the range of $\sim 10^9$ M$^{-1}$ s$^{-1}$. The similar results can be drawn for AS-GL
mixture, suggesting that the electrostatic attraction is a significant factor to affect the NA reactions.
To further evaluate the cationic reaction mechanism, the natural bond orbital (NBO) analysis reveals that the N atom of
$NH_3$ exhibits more negative charge ($-1.1$ e) relative to MEA ($-0.9$ e), suggesting the stronger electrostatic attraction between
$CB_{DL}$ and $NH_3$ to yield $CI_{AS}1$ in the first-step NA reaction (see NA1a and NA2a in Fig. 4). However, the second-step NA
reaction between $CB_{MEA}$ and MEA are promoted by MEA because the presence of MEA enhances the positive charge in $CB_{MEA}$
(0.6 e), facilitating the electrostatic attraction (see NA1b and NA2b in Fig. 4). For the third-step NA reaction (see NA1c and
NA2c in Fig. 4), due to the steric hindrance, the deprotonation of $CI_{MEA}8$ possesses a larger $\Delta G^{\ddagger}$ value relative to that of $CI_{AS}8$.
Hence, the NA reactions are regulated by both electrostatic attraction and steric hindrance effect.
The fourth-step NA reactions in MEA-GL and AS-GL mixtures exhibit two distinct chemical reaction mechanisms in
cyclization to yield N-heterocycles (see NA1d and NA2d in Fig. 4). The protonation of $PIC_{MEA}$ and $PIC_{AS}$ occurs at the
hydroxyl group to form $CI_{MEA}9$ and $CI_{AS}9$. For MEA-GL mixture, the barrierless dehydration and cyclization of $CI_{MEA}9$ occur
in one step to yield N-heterocycle (i.e., $IC_{MEA}$), with the total $\Delta G_r$ value of $-78.6$ kcal mol$^{-1}$ (NA1d in Fig. 4a). However, for
AS-GL mixture, the cyclization of $PIC_{AS}$ to $IC_{AS}$ includes protonation, dehydration, cyclization, and deprotonation. Note that
cyclization and deprotonation proceed via two TSs in sequence, with the corresponding $\Delta G^{\ddagger}$ values of 3.9 and $-0.6$ kcal mol$^{-1}$
(NA2d in Fig. 4b), respectively, forming $IC_{AS}$. As discussed above, cyclization in MEA-GL and AS-GL mixtures is the rate-
limiting step to chromophore formation.
For MEA-AS-GL mixture, $AHA_{MEA/AS}$, and $ID_{MEA/AS}$ are yielded via the same first NA reactions (NA1a/2a) as MEA-GL
and AS-GL mixtures. Also, the formation of $ID_{MEA/AS}$ proceeds via protonation and dehydration to form $CB_{MEA/AS}$. However,
the second NA reaction includes the cross-NA reaction of $CB_{MEA}$ with AS (NA3b-1) or $CB_{AS}$ with MEA (NA3b-2) to produce
extra oligomers (i.e., $HA_{MAG}1$ and $HA_{MAG}2$), in contrast to MEA-GL and AS-GL mixtures. Hence, the fate of $CB_{MEA/AS}$ is





dependent of the competition reaction between the pathways of self-NA reaction to form HA $_{MEA/AS}$ (NA1b/2b) and cross-NA
reaction to yield HA$_{MAG}$1/2 (NA3b-1/2). The $\Delta G_r$ values of the cross-NA reactions to yield HA$_{MAG}$1 and HA$_{MAG}$2 are −30.3
and −30.4 kcal mol$^{-1}$, respectively, comparable with those of self-NA reactions. It suggests both NA reactions to form HAs are
equally accessible. Subsequently, HA$_{MAG}$1/2 undergoes dehydration to form DI$_{MAG}$, further proceeds the third NA reaction to
yield PIC$_{MAG}$, in line with the mechanisms of the third NA reactions for MEA-GL and AS-GL mixtures. The cyclization of
CI$_{MAG}$10 (the fourth NA reaction) possesses with two successive TSs, similar to that of AS-GL mixture but different to that of
MEA-GL mixture. The corresponding $\Delta G^{\ddagger}$ values are obtained as 5.0 and 1.6 kcal mol$^{-1}$, respectively, which are larger than
those of AS-GL mixture. In summary, compared with the AS-containing mixtures, the presence of MEA provides the extra two
branched chains in N atoms, which affect the natural charges and molecular steric hindrance of intermediates, to thereby
facilitate the intramolecular interaction between N and C atoms to form SBrC chromophores.

## 4 Conclusions and atmospheric implications

BrC chromophores play an important role in the Earth's radiative balance, air quality and human health. However, the
formation mechanisms of BrC chromophores are not fully understood, hindering a comprehensive assessment of BrC
chromophores on atmospheric chemistry and environmental impacts. Hence, using combined theoretical and experimental
methods, we investigated the aqueous chemistry of typical RNCs with GL and evaluated the impact of typical multifunctional
RNCs on the formation of BrC chromophores. Experimental studies show that the MAC values of chromophores are affected
by the initial pH value for AS-GL, MEA-GL and MEA-AS-GL mixtures, and the growth rates of chromophores are enhanced
in the presence of MEA. The optical properties of chromophores are regulated by monocyclic and bicyclic IML compounds in
AS-GL mixture but by monocyclic IML compounds in MEA-containing mixtures (i.e., MEA-GL and MEA-AS-GL).
Combined with the results of quantum chemical calculations, chromophore formation is characterized by nucleophilic addition
with large exothermicity and strong electrostatic attraction among the MEA-derived intermediates, which are also enhanced
by MEA.

283       In addition, to simply evaluate the impacts of MEA and AS on chromosphere formation in the aqueous aerosols and

fog/cloud droplets, the dynamics process of GL from gas to aqueous phase was carried out (Fig. S12). The free energy
difference reflects whether the liquid particles with MEA and AS (denoted as MEA and AS particles) prefer to adsorb and
accommodate GL. As shown in Fig. S12, a larger decrease in the free energy (−3.7 kcal mol$^{-1}$) occurs when GL approaches
the interface of the MEA particle relative to the AS particle, indicating a thermodynamically favorable process. Subsequently,
the stabilized GL enters into the interior region of the MEA and AS particles, with slightly endothermic (1.6 and 2.4 kcal mol$^{-}$
$^{1}$). A smaller free energy difference from the interface into the interior region of the MEA particle implies that the interfacial
GL is more readily promoted to enter the interior region of the particle when the particles contain MEA compared with AS.
Hence, the interfacial and interior attraction on the MEA particle is more pronounced for small α-dicarbonyls, to facilitate the
further engagement in the aqueous-phase reactions with RNCs in the particle.



Formation of SBrC from multifunctional RNCs and small α-dicarbonyls occurs widely on aqueous aerosols and fog/cloud
droplets under typical atmospheric conditions. Compared with the ubiquitous coexistence between AS and small α-dicarbonyls
from global aerosol measurement, SBrC aerosol formation from multifunctional RNC mixtures should be paid attention to
during serious haze formation in China because of their atmospheric reactivities and non-negligible concentrations. Our results
also imply that SBrC aerosols, if formed from the aqueous reactions between MEA and GL, likely contribute to atmospheric
warming because the presence of MEA enhances the MACs of the mixture. Hence, recognition of this aerosol formation
mechanism in the radiative transfer atmospheric model is needed, reparenting a possible missing source for BrC formation on
urban, regional and global scales.
**Data availability.** All raw data can be provided by the corresponding authors upon request.
**Supplement.** The supplement related to this article is available on the EGU Publications website.
**Author contributions.** YMJ and ZS designed the research; YMJ, ZS, RZM, and WJL performed the research; YJ, ZS, WJL,
JXW, QJS, YXL, LG, LLX, YPG, GYL, and TCA analyzed the data; YMJ and ZS wrote the paper. YMJ, YXL, YPG, GYL,
and TCA reviewed and edited the paper.
**Competing interests.** The contact author has declared that neither they nor their co-authors have any competing interests.
**Disclaimer.** Publisher's note: Copernicus Publications remains neutral with regard to jurisdictional claims in published maps
and institutional affiliations.
**Financial support.** This work was financially supported by National Natural Science Foundation of China (grant nos.
42077189 and 42020104001), Guangdong Basic and Applied Basic Research Foundation (Grant Nos. 2019B151502064),
Local Innovative and Research Teams Project of Guangdong Pearl River Talents Program (Grant Nos. 2017BT01Z032), and
Guangdong Provincial Key R&D Program (Grant Nos. 2022-GDUT-A0007).
**Review statement.** This paper was edited and reviewed by two anonymous referees.

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






Figure 1: The MAC values for AS-GL, MEA-GL and MEA-AS-GL mixtures at the initial pH of 3 and 4 at 1d (a) and 15d (b).







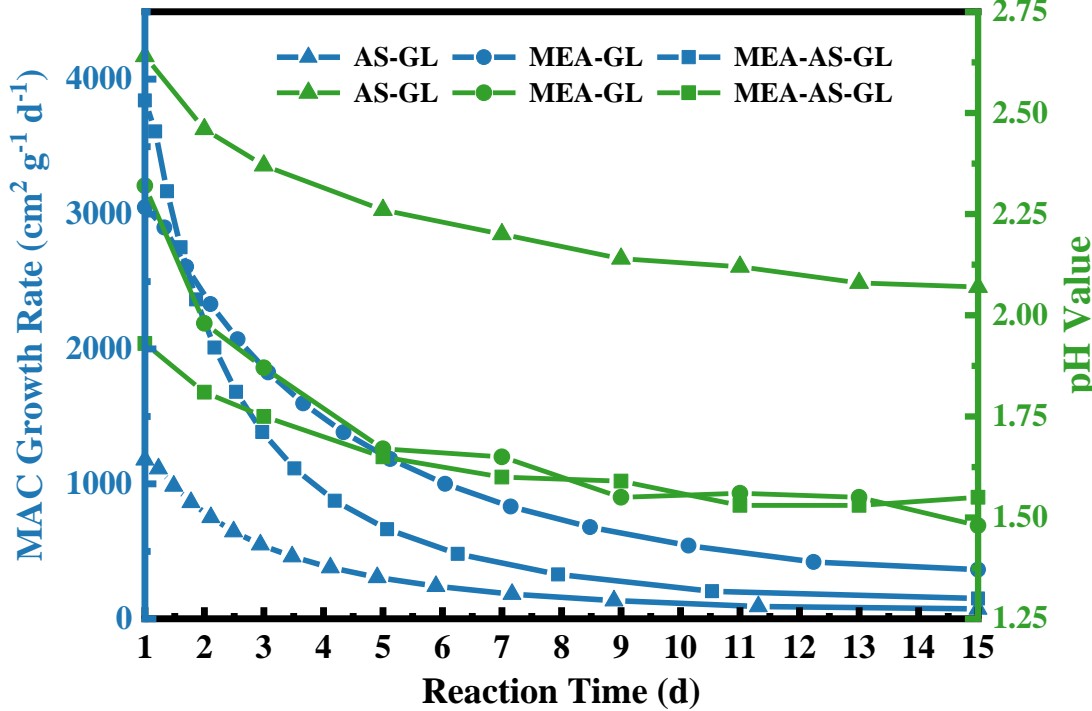


**Figure 2: Dependence of the growth rates (blue line) and pH values (green line) on reaction time for AS-GL, MEA-GL and MEA-**
**AS-GL mixtures.**





**Figure 3: Mass spectra monitoring of chromophores for (a) MEA-GL (b) AS-GL and (c) MEA-AS-GL mixtures.**



**Figure 4: Possible pathways leading to chromophores for (a) MEA-GL and (b) AS-GL mixtures (oriented by gray arrows). Detailed PESs of the four NA reactions are presented in dotted boxes. The number denotes the values of $\Delta G_r$ and $\Delta G^{\ddagger}$ (in brackets) for each reaction step (in kcal mol$^{-1}$), and all energies are relative to the corresponding reactants.**



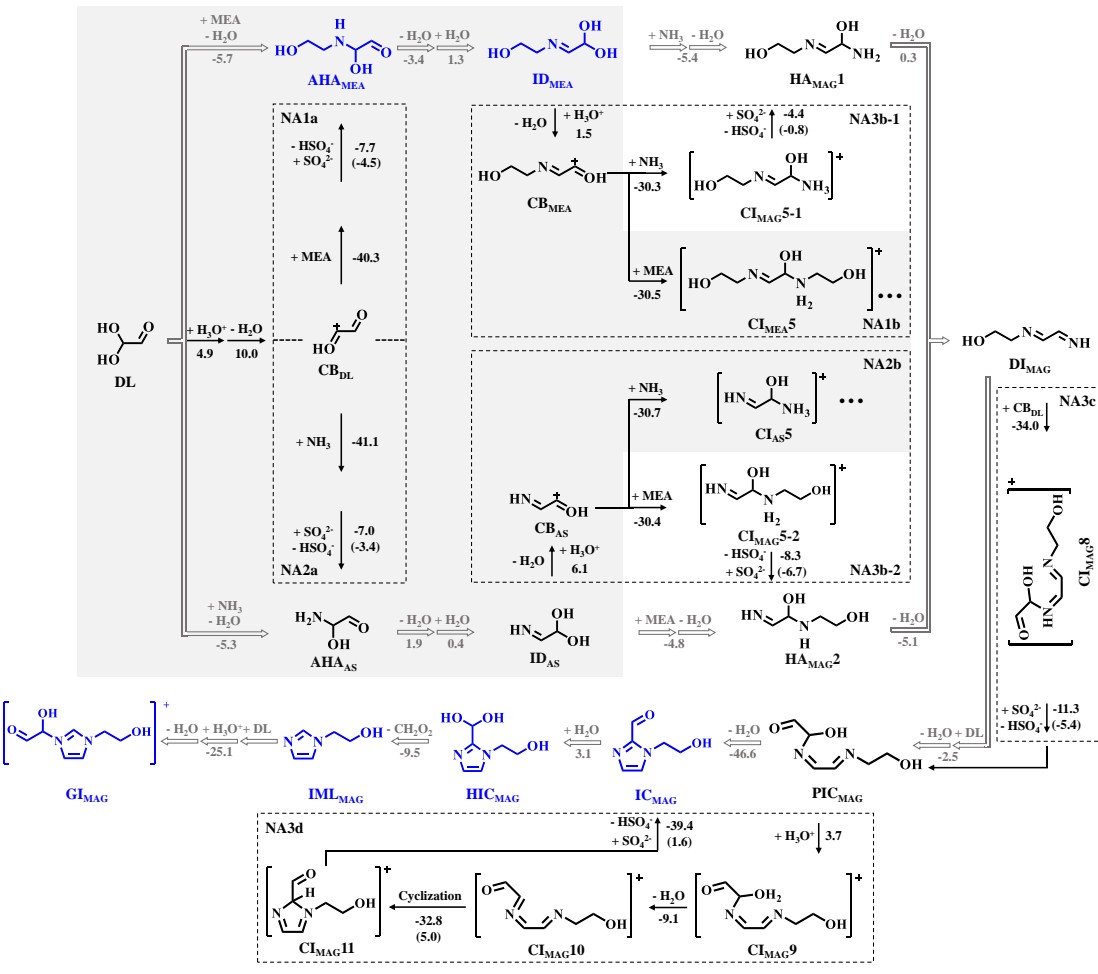

541

**Figure 5: Possible pathways leading to chromophores for MEA-AS-GL mixture (oriented by gray arrows). Detailed PESs of the four NA reactions are presented in dotted box. The shaded area is the overlapping part with the pathways of MEA-GL and AS-GL mixtures. The number denotes the value of $\Delta G_r$ and $\Delta G^{\ddagger}$ (in brackets) for each reaction step (in kcal mol$^{-1}$), and all energies are relative to the corresponding reactants.**

546