# Peer review of "Aqueous-phase chemistry of glyoxal with multifunctional reduced"

_EGUsphere, 2023_

## Author Comment (AC1)

**Manuscript ID: egusphere-2023-2662**

**Title:** Aqueous-phase chemistry of glyoxal with multifunctional reduced nitrogen compound: A potential missing route of secondary brown carbon

**The corresponding authors:** Prof. Taicheng An

**Dear Anonymous Referee #1,**

Thank you for the helpful and valuable review and comment. We have made careful revisions on the original manuscript according to your kind and helpful comments. The changed sentences have been marked as red color in the revised manuscript. Below is our point-by-point response to your comments:

**Question 1**. The authors presented experimental results of aqueous-phase (dark? Please specify measures to avoid photolysis of the light-absorbing product formed or light-induced reactions, if any) reactions of glyoxal (GL) and two types of reduced nitrogenous species (ammonium sulfate, AS, and monoethanolamine, MEA).

**Response:** We are sorry for causing you confusion due to our unclear expression. In this study, we used the brown vials to place the mixtures, which have been proven to can effectively avoid the photolysis and light-induced reactions of light-absorbing products (*Atmos. Chem. Phys.*, 2012, 12(14), 6323-6333). According to the reviewer's helpful suggestion, we have modified the relevant sentences in the revised manuscript: "**Each mixture was transported into brown vials, which have been proven to avoid the photolysis and light-induced reactions of light-absorbing products (Kampf et al., 2012), to guarantee efficiently produce chromophores in droplet evaporation collecting on the timescales of seconds (Zhao et al., 2015; Lee et al., 2014).**" (Please see lines 80-83)

Reference:
Kampf, C. J., Jakob, R., and Hoffmann, T.: Identification and characterization of aging products in the glyoxal/ammonium sulfate system - implications for light-absorbing material in atmospheric aerosols, Atmos. Chem. Phys., 12, 6323-6333, https://doi.org/10.5194/acp-12-6323-2012, 2012.

**Question** 2. L131: the maximum peak absorption of 207 - 212 nm might not be of concern as for light absorption in the troposphere (and they might not make their way to the stratosphere). The MAC values reported here ($10^3$ - $10^4$ cm$^2$ g$^{-1}$) were compared to those in literature, but most previous studies focused on the MAC values of

tropospherically relevant peaks of 290 - 320 nm (the minor one). I just randomly checked one that was cited by the authors, they reported MAC values in this wavelength range (~290 - 320 nm) to be on the order of $10^4$ cm$^2$ g$^{-1}$, similar to the range reported by the authors of the current study for 207 - 212 nm. Therefore, I do not find this comparison very convincing.

**Response:** Thanks for the reviewer's comment. In our study, two adsorption peaks are observed in the wavelength range of 285 - 324 nm and 207 - 212 nm, with a consistent trend. In this section, our aims to discuss and compare the differences of the absorbance in three target reaction systems and the effect of the initial pH on reaction systems. Hence, to better show the changes, the maximum absorption peaks in the range of 207 - 212 nm are selected and discussed in this study. In addition, note that the MAC values in the reference randomly checked by the reviewer are most obvious at 290 - 320 nm because of the longer reaction time of 2 - 3 months than ours (15 days), while the MAC values in the other literature we cited with a reaction time of 4 days are similar with our values. It indicates that the MAC values of the maximum absorption peaks in the range of 290 - 320 nm increase significantly with the increase of reaction time. After the reviewer's kind reminder and suggestion, to make the article more rigorous, we have added the relevant discussions and Fig. S4 and have updated the references in the revised manuscript:
**"In addition, each of the mixtures has an absorption peak between 285 - 324 nm (Fig. S4), with a range of 42 - 228 cm$^2$ g$^{-1}$, which are consistent with the MAC values measured by Powelson et al. at the reaction time of 4 days (Powelson et al., 2014) but are smaller than the values measured by Zhao et al. with the long reaction time of 2 - 3 months (Zhao et al., 2015). The MAC values at 207 - 212 and 285 - 324 nm exhibit a similar trend (Fig. S4). Therefore, to easily compare the absorbance in three mixtures, we focus on the adsorption peaks in the range of 207 - 212 nm, which exhibits an obvious variation, and the effect of the initial pH on the reaction systems is also discussed in this range."** (Please see lines 134-140)

References:
Powelson, M. H., Espelien, B. M., Hawkins, L. N., Galloway, M. M., and De Haan, D. O.: Brown carbon formation by aqueous-phase carbonyl compound reactions with amines and ammonium sulfate, Environ. Sci. Technol., 48, 985-993, https://doi.org/10.1021/es4038325, 2014.

Zhao, R., Lee, A. K. Y., Huang, L., Li, X., Yang, F., and Abbatt, J. P. D.: Photochemical processing of aqueous atmospheric brown carbon, Atmos. Chem. Phys., 15, 6087-6100, https://doi.org/10.5194/acp-15-6087-2015, 2015.

[Figure]

**Figure. S4.** The MAC values for AS-GL (a and b), MEA-GL (c and d) and MEA-AS-GL (e and f) mixtures at the initial pH of 3 and 4 in the wavelength range of 250 - 350 nm over a time scale of 15 d.

**Question** 3. I find it quite interesting, but puzzling, that the pH values decreased by as much as 2.5 units. What was the acidic species formed after 15 days that might contribute to more than 2 orders of magnitude increase in proton concentration (or more accurately activity)? If that is the case (pH decreased by maximum 2.5 units), would the light absorption of the products be affected by such dramatic pH change as the amine groups got more protonated?

**Response:** We are grateful for the feedback provided by the reviewer. According to our experimental and theoretical results, it is formic acid leading to a decrease of 2.5 units in the pH value, in line with the results and analysis from the previous studies (*J. Org. Chem.*, 1995, 60, 6246-6247; *Environ. Sci. Technol.*, 2011, 45, 6336-6342).

For example, Yu et al. conducted a study on the dark reaction of AS-GL using $^1$H NMR, and the results have revealed that the formation of formic acid is the primary reason for the rapid decrease in the pH value from 4.4 to about 3.4 in AS-GL mixture. In addition, our theoretical calculations also show that the products HIC$_{MEA}$, HIC$_{AS}$, and HIC$_{MAG}$ can undergo C–C bond cleavage, forming formic acid, with the reaction energies from -9.5 to -15.6 kcal mol$^{-1}$. It indicates that the formation of formic acid is thermodynamically favorable, in consistent with the results obtained by Kua et al. that imidazole materials generated by methylamine-GL can also undergo C–C bond cleavage to form formic acid (*J. Phys. Chem. A*, 2011, 115, 1667-1675). In addition, the decrease in pH value has little effect on the optical properties of the absorbent products because (i) the absorbance of light-absorbing products such as imidazole compounds is mainly contributed by the n→π* and π→π* transitions of the conjugated system composed of C=N and C=C in the imidazole ring. When protonation of the amine groups in the light-absorbing products occurs, the structures of C=N and C=C are not affected, implying that there is a little impact on the absorbance of the light-absorbing products. (ii) the formation of formic acid will lead to the decrease of pH value in the solution environment, thereby reducing the rate of the nucleophile addition reaction but not affecting its mechanism. After reviewing our existing theoretical calculation and experimental results, we have confirmed that the reaction mechanism and types of the reaction products remain unchanged. According to the reviewer's comments, some discussions are revised and added in the revised manuscript: **"Similarly, IC$_{MEA}$ also undergoes a hydration reaction to form HIC$_{MEA}$ with a similar structure to HIC$_{AS}$. Subsequently, HIC$_{AS}$ and HIC$_{MEA}$ are decomposed to yield IML$_{AS}$ and IML$_{MEA}$, respectively, accompanied by the formation of formic acid ($\Delta G_r$ = -10.2 and -15.6 kcal mol$^{-1}$), which is the reason for the decrease in pH in Section 3.1. However, as a reaction byproduct, formic acid hardly participates in the formation of light-absorbing products, so it has little influence on the reaction mechanisms."** (Please see lines 200-204)

**Question** 4. Maybe I missed it somewhere, but is Figure 3 showing "ensemble" mass spectra of many peaks (products) in LC chromatograms, or a mass spectrum of an individual product? There is no clear description on how well the products were separated in LC.

**Response:** We thank the reviewer for this constructive comment. The original Figure. 3 was the overall mass spectra of all the products identified in each mixture. According to the reviewer's suggestion, we have supplemented Figs. S5 - S7 with the extracted ion chromatograms for each major product.

[Figure]

**Figure S5.** The extracted ion chromatograms of all reaction products for MEA-GL mixture at the initial pH of 3 and 4.

[Figure]

**Figure S6.** The extracted ion chromatograms of all reaction products for AS-GL mixture at the initial pH of 3 and 4.

[Figure]

**Figure S7.** The extracted ion chromatograms of all reaction products for AS-MEA-GL mixture at the initial pH of 3 and 4.

**Question** 5-1. I am not sure how much the calculation of diffusion-controlled rate constant (page S1-S2 in the SI) is going to affect the kinetic analysis.

**Response**: We would like to express our gratitude for the reviewer's comment. The diffusion-controlled rate constant is a key parameter in liquid-phase reaction kinetics. More importantly, it has a significant impact on the apparent rate constant when the

intrinsic reaction rate constant is larger than $10^9$ $M^{-1}$ $s^{-1}$ (*Chem. Rev.*, 1999, 99, 2161-2200; *Environ Int*, 2019, 129, 68-75). It is also an important feature that distinguishes liquid-phase reaction kinetics from gas-phase reaction kinetics. According to the previous studies (*Water Res.*, 2014, 49, 360-370; *Proc. Natl. Acad. Sci. U.S.A*, 2020, 117, 13294-13299), the barrierless bimolecular reactions are mainly determined by the diffusion ability of the molecules in the liquid phase, of which the rate constants are approximately equal to the diffusion-controlled rate constants; for the bimolecular reactions with transition states, the diffusion effect should be considered if the rate constants are larger than $10^9$ $M^{-1}$ $s^{-1}$.

**Question** 5-2. But I have a few questions about Eq. 5 on Page S2: 1) why was the viscosity of the solvent used, given a relatively high concentration (approximately 1 M), and high-molecular-weight products were formed (high MW, higher viscosity);

**Response**: In this work, the viscosity of solvent (i.e., water) was applied for the following reasons: (i) at 1 M concentration, GL and RNCs still serve as solutes (i.e., reactants) to undergo the liquid-phase reactions, which are affected by the viscosity of water; (ii) the MWs of the imidazole-like products are less than 200, and thereby, there is a little influence of the viscosity by the products from reactions we focused on; (iii) based on the previous studies, the viscosity varies from 0.89 mPa s (0 wt%, pure water) to 1.058 mPa s in a 10 wt% imidazole solution (*J. Chem. Eng. Data*, 2019, 64, 2, 507–516), and the concentration of imidazole-like products is considerably below 0.4 wt% (*Environ. Sci. Technol.*, 2011, 45, 6336-6342). It suggests that the viscosity of solvent is not affected by the formation of high MW products (i.e., imidazole-like products) in this study because of the small concentration of the imidazole-like products; (iv) the viscosity of water is also used to evaluate the kinetics of elementary reactions by other previous studies (*Water Res.*, 2014, 49, 360-370; *Org. Lett.*, 2009, 11, 22, 5114-5117; *Proc. Natl. Acad. Sci. U.S.A*, 2020, 117, 13294-13299; *Atmos. Chem. Phys.*, 2022, 22, 7259-7271). Hence, the diffusion-limit effect is assessed by using the viscosity of water in our study.

**Question** 5-3. Is fractional S-E equation needed for the estimation of diffusion coefficient (it is widely known that the S-E equation does not work very well for small species);

**Response**: Some studies have pointed out that the diffusion coefficients ($D$) based on the S-E equation exhibit the discrepancies from experimental values, particularly when dealing with small species, because there is an uncertainty in the assessment of the particle size for small species. Hence, some efforts have been made to improve the accuracy of S-E results, such as incorporating correction terms or refining the calculation of radii, to fit the small species systems (*Angew. Chem. Int. Ed.*, 2013, 52, 3199-3202). According to the reviewer's helpful suggestion, we evaluate the impact of the radii under the original S-E and the modified S-E equation. The results revealed that our calculated $D$ values agree with those obtained through the correction method mentioned above, with the errors within the range of $(1.44{\sim}3.48)$ $\times 10^{-10}$ m$^2$ s$^{-1}$. Therefore, the diffusion-controlled rate constants derived from the $D$ values by both methods also agree with each other, in the same order of magnitude ($\times 10^9$ M$^{-1}$ s$^{-1}$). Furthermore, the original S-E aquation has been successfully applied in recent studies on the evaluation of the diffusion coefficients in small species reaction systems (*Water Res.*, 2014, 49, 360-370; *Ecotoxicol. Environ. Saf.*, 2022, 231, 113179.; *Proc. Natl. Acad. Sci. U.S.A*, 2020, 117, 13294-13299; *Atmos. Chem. Phys.*, 2022, 22, 7259-7271). It indicates that the S-E equation is suitable to assess the rate constants in our study.

**Question** 5-4. what parameterization was used to calculate the mutual diffusion coefficient of D_AB from D_A and D_B?

**Response**: According to the reviewer's helpful suggestion, the calculation details of $D_A$, $D_B$ and $D_{AB}$ have been revised to the "Detailed description of rate constants" in SI: **"$D_{AB}$ is the mutual diffusion coefficient of the reactants A and B, which is the sum of diffusion coefficients of reactants A and B ($D_A$ and $D_B$), i.e., $D_{AB} = D_A + D_B$ (Truhlar, 1985). $D_A$ and $D_B$ are estimated from the Stokes–Einstein approach (Einstein, 1905) listed in expression (5):**

$$D = \frac{k_B T}{6\pi\eta a} \tag{5}"$$

**Technical:**

**Question** 1. P3/L67: "amines". You only have results for one amine, right?

**Question** 2. P3/L86: "spectrums" to "spectra"?

**Question** 3. P6/L164: "were" to "was"?

**Response:** According to the reviewer's suggestion, we have revised the corresponding sentences according to the reviewer's suggestion. (Please see lines 67, 68, 87, and 168)

---

## Author Comment (AC2)

**Manuscript ID: egusphere-2023-2662**

**Title:** Aqueous-phase chemistry of glyoxal with multifunctional reduced nitrogen compound: A potential missing route of secondary brown carbon

**The corresponding authors:** Prof. Taicheng An

**Dear Anonymous Referee #2,**

Thank you for the helpful and valuable review and comment. We have made careful revisions on the original manuscript according to your kind and helpful comments. The changed sentences have been marked as red color in the revised manuscript. Below is our point-by-point response to your comments:

**Major Comments:**

**Question** 1. The manuscript discussed about the effects of pH on the reaction, which seems to be contradictory. First of all, lines 140-154, the experimental data show that higher pH will lead to higher MAC values and more SBrC formation.

However, Figure 2 shows that the pH drop for MEA-GL mixture is faster and MEA-GL showed lower pH values than the other two mixtures. The author also stated in lines 160-162 that lower pH will lead to more SBrC formation.

These two statements in the manuscript are self-contradictory and should be addressed and reconciled.

**Response:** Thank you for the referee's comment. There is no contradiction between the results presented in lines 140 - 154 and 160 - 162 because the pH values utilized belong to distinct categories. The results of "higher pH will lead to higher MAC values and more SBrC formation" from lines 140 - 154 are obtained by using the initial pH values. That is, the chromophore formation rates and MAC values of the RNCs-GL reaction mixtures are increased when the initial pH values are increased. As the reaction goes on, the ambient pH values are decreased due to the formation of formic acid as a byproduct, which has been discussed and obtained in Sections 3.2 and 3.3. Hence, the results of "lower pH will lead to more SBrC formation" are deduced. We are very sorry for the confusion caused by our unclear expression. According to the referee's comments, some discussions are revised and added to the revised manuscript: **"In order to explore the influence of the initial pH values on the MAC values, a comparison of MAC values at initial pH 3 and 4 is performed for all three mixtures (Fig. 1a)."** (Please see lines 140 - 141) and

**"Note that the trend of GRs shows a decrease from MEA-AS-GL, MEA-GL, to AS-GL mixtures at the beginning of the reaction time, while the MAC values of MEA-GL mixture are larger than those of two mixtures accompanied by the more rapid decrease of pH values in solution after the reaction is equilibrium (Figs. 1b and 2), suggesting that chromophore formation of three mixtures depends on the ambient pH value."** (Please see lines 161 - 164)

**Question** 2. Lines 189-240, the quantum calculations show that the ΔG values for the reaction of all three mixtures to generate DI$_{MEA}$ are: MEA-GL<AS-GL, MEA-AS-GL>AS-GL on day 15. It is strange that the ΔG value shows AS-GL<MEA-AS-GL, while the MAC value shows the opposite trend where MEA-AS-GL>AS-GL.

The authors should explain further why these is this contradiction between the experimental value and quantum modeling results.

**Response:** We thank the referee for this constructive comment. The theoretical predictions are coincident with the experiment results because of the following reasons: (i) as discussed in section 3.2, we focus on the feasibility of the formation of intermediates rather than the order of the difficulty. Therefore, we utilized the reaction energies ($\Delta G_r$) rather than the activation energies ($\Delta G^{\ddagger}$) because the $\Delta G_r$ values are good at predicting the possibility of the formation of intermediates; (ii) the $\Delta G_r$ values mentioned by the referee are only $\Delta G_r$ values for the formation of the intermediates, i.e., DI$_{MEA}$, DI$_{AS}$, and DI$_{MAG}$. It only indicates that the formation of DI$_{AS}$ from HA$_{AS}$ is thermodynamically more feasible than the formation of DI$_{MAG}$ from HA$_{MAG}$1, rather than specifying the formation of chromophores is easier. The total $\Delta G_r$ value of AS + GL → DI$_{AS}$ is -7.8 kcal mol$^{-1}$, which is 5.1 and 10.9 kcal mol$^{-1}$ higher than the $\Delta G_r$ value of MEA + AS + GL → DI$_{MAG}$ (-12.9 kcal mol$^{-1}$) and MEA + GL → DI$_{MEA}$ (-18.7 kcal mol$^{-1}$), respectively, indicating that the formation of DI is thermodynamically most feasible in the MEA-GL mixture, in agreement with the highest MAC value in MEA-GL mixture measured in the experiment; (iii) in section 3.3, we focus on the mechanisms for the formation of chromophores in three mixtures, a detailed potential energy surface of the key elementary reactions predicted in section 3.2 was calculated and established using the activation energies ($\Delta G^{\ddagger}$). Through the discussion of reaction barriers combined with geometries and natural charges, we found that the nucleophilic

addition (NA) reactions are regulated by both electrostatic attraction and steric hindrance effect. The additional branched chain on the N atom in MEA affected the natural charge and steric hindrance of the reaction intermediate, thus promoting the intramolecular interaction between N and C atoms to form SBrC chromophore, thus causing the MEA-AS-GL mixture to have a higher MAC value than AS-GL mixture.

**Minor comments**

**Question** 1. Line 79: when altering the pH of the solution with addition of sulfuric acid or sodium hydroxide solution, will this also dilute the solution and cause the concentration of each solution to be different and diverge from 1M?

**Response:** Thank you for the questions raised by the referee. It will not affect the concentration of the mixture because the strongly alkaline MEA was pre-acidified before the mixing process and is followed by precise dilutions using a volumetric flask. In the process of preparing the mixture, the strongly alkaline MEA was pre-acidified by sulfuric acid first to ensure that the pH value of the mixture after mixing is between 3 and 4, which is our target pH value. After mixing and diluting, each mixture was fine-tuned with sulfuric acid or sodium hydroxide to reach a pH of 3 or 4. In this pH adjustment process, the sulfuric acid or sodium hydroxide solution we add is less than 1mL, and the concentration change is less than 2%. In summary, the use of sulfuric acid or sodium hydroxide solution to adjust the pH value has little effect on the concentration of each mixture.

**Question** 2. The word "spectrums" should be changed "spectra" throughout the text and also in the SI. For instance, Figures S4-S8 used spectrums.

**Response:** According to the referee's suggestion, we have changed all "spectrums" to "spectra" in the text and SI.

**Question** 3. For all MS spectra (Figures 3, S4-S8), I suggest the authors should also show chromatographs as well.

**Response:** Thank you for the referee's comment. We have added extracted ion chromatograms of all reaction products of into Figures S4 - S6 in the SI.

[Figure]

**Figure S4.** The extracted ion chromatograms of all reaction products for MEA-GL mixture at the initial pH of 3 and 4.

[Figure]

**Figure S5.** The extracted ion chromatograms of all reaction products for AS-GL mixture at the initial pH of 3 and 4.

[Figure]

**Figure S6.** The extracted ion chromatograms of all reaction products for AS-MEA-GL mixture at the initial pH of 3 and 4.